# Addressing the Cold-Start Problem in Recommender Systems Based on Frequent Patterns

Antiopi Panteli [ID] and Basilis Boutsinas *[ID]

Management Information Systems & Business Intelligence Laboratory, Department of Business Administration, University of Patras, GR 26504 Patras, Greece
* Correspondence: vutsinas@upatras.gr

**Abstract:** Recommender systems aim to forecast users' rank, interests, and preferences in specific products and recommend them to a user for purchase. Collaborative filtering is the most popular approach, where the user's past purchase behavior consists of the user's feedback. One of the most challenging problems in collaborative filtering is handling users whose previous item purchase behavior is unknown, (e.g., new users) or products for which user interactions are not available, (e.g., new products). In this work, we address the cold-start problem in recommender systems based on frequent patterns which are highly frequent in one set of users, but less frequent or infrequent in other sets of users. Such discriminant frequent patterns can distinguish one target set of users from all other sets. The proposed methodology, first forms different clusters of old users and then discovers discriminant frequent patterns for each different such cluster of users and finally exploits the latter to hallucinate the purchase behavior of new users. We also present empirical results to demonstrate the efficiency and accuracy of the proposed methodology.

**Keywords:** cold-start problem; sparsity; recommender systems; discriminant frequent patterns

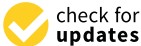



## 1. Introduction

Recommender systems aim to forecast a user's rank, interests, and preferences in specific products (or services) and make corresponding recommendations to the user for purchase. The recommender systems forecast product purchases from a very large inventory of products based on the user's past purchase behavior and possibly on the user's current context. The demand for recommender systems has arisen because of the information surplus created through the huge volume of data.

Recommender systems can be broadly divided into collaborative filtering-based, content-based, and hybrid systems [1]. Content-based filtering (CBF) exploits product characteristics and user demographics to recommend similar products to the ones that the user purchased in the past. Collaborative filtering (CF) is the most popular approach, and it utilizes the opinions of other users with similar preferences, given that similarities can be computed from the historical transactions of the user under observation.

Collaborative filtering can be divided into two categories, i.e., memory-based and model-based. Memory-based or user-based takes into consideration previous user product ratings to recommend products to a user for purchase based on similarity measures and correlation. Model-based collaborative filtering predicts rates of products to be recommended to the user through the development of a model with the use of machine learning techniques such as clustering, neural networks, decision trees, etc. In the rest of the paper, we use the term product to also denote services and information items.

In collaborative filtering, the user's past purchase behavior consists of the user's feedback which can be either implicit or explicit [2]. Implicit feedback is usually binary and it is obtained by observing the user's behavior (products purchased or not, songs listened or skipped, web pages browsed or not, content downloaded or not, etc.) [3]. Explicit

feedback is the discrete ratings assigned by the user for every product purchased. Users and products are divided into warm (users whose previous product purchase behavior is known or products for which user interactions are available) and cold (users whose previous product purchase behavior is unknown, e.g., new users or products for which user interactions are not available, e.g., new products). Recommendations concerning the latter constitute the popularly known cold-start recommendation problem.

When CF faces the cold-start problem, it is difficult to provide useful recommendations to the user, since no prior ratings exist. Therefore, content-based filtering (CBF) is used instead to provide recommendations. However, CBF algorithms recommend products based on similar content. This leads to recommendations that are less diverse with each other. The latter poses various problems, e.g., when a user considers one recommendation as useless, it is highly possible that all the rest similar recommendations might be considered worthless as well.

In this paper, we address the cold-start problem in recommender systems by applying both clustering and association rule mining (ARM) techniques. More specifically, the proposed methodology is based on discriminative itemset mining technique that extracts discriminant frequent patterns for each different cluster of old users extracted by a clustering technique. To our knowledge, the proposed methodology is the first one that exploits discriminative itemset mining to address the cold-start problem for both its *in-matrix prediction* and *out-of-matrix prediction* setup. *In-matrix prediction* refers to users that purchased at least one product (or to products that have been rated by at least one user). The proposed methodology addresses also the *out-of-matrix prediction*, where new users have never carried out any purchase before (or products that have never been rated before). Thus, the proposed methodology tackles a more challenging problem that is not often studied in the literature [4] and fills this research gap. With respect to similar research, the main contributions of this work are the following:

1. It proposes a novel hybrid approach, which combines clustering, discriminative itemset mining and the user's/product's context.
2. It can handle both the sparsity and the cold-start recommendation problems.
3. It can handle the *out-of-matrix prediction* (pure) cold-start recommendation problem, where users have no rating history. In most of similar approaches, several initial ratings are needed.
4. It does not need any interview process to ask the user to fill in some extra information (e.g., extended profile data, answer questions and rate items), before using the system for the first time [5].
5. It does not include many modules that are interrelated with multiple dependencies.

Discriminative itemset mining is a focused association rule mining research area for discovering interesting patterns that state the significant differences between datasets [6,7]. The discriminative itemset mining across multiple datasets captures the itemsets which are highly frequent in one dataset but less frequent or infrequent in other datasets.

We assume an existing recommender system with an established set of warm users and products. The proposed methodology first forms different clusters of warm users and then discovers the discriminant frequent patterns for each such different cluster of users and finally exploits the latter to hallucinate the purchase behavior of cold users. In what follows, first, we present related work and then we analyze the proposed methodology. We also present empirical results to demonstrate the efficiency and accuracy of the proposed methodology. We apply the proposed methodology to the MovieLens dataset which has already been widely used as a benchmark for recommender systems [8–14].

## 2. Literature Review

The approaches proposed in the literature, addressing the cold-start recommendation problem by exploiting implicit information, can be broadly grouped into three categories, i.e., the similarity-based models, matrix factorization models, and feature mapping models [4]. Similarity-based models aim at predicting user–product matrix for the cold users by assigning a cold user to a group, based on usually applying machine learning techniques to both implicit and explicit information and then on detecting similar warm users. Matrix factorization and feature mapping models factorize the user–product matrix into two latent representations. Then, one can learn a latent representation using the Implicit information, and then use it to predict the user–product matrix for the cold users.

For instance, a similarity-based model presented in [15], where a generalized matrix algebra framework is defined using generalized matrix multiplication permitting any similarity metric operators, e.g., inner product and cosine similarity. This framework is applied to a matrix of user–product purchases where the purchased products are viewed as user attributes and extends this matrix with non-product personal attributes P, e.g., demographics, Facebook friends, or page likes. As another example, in [16] classification is used to assign a new user to a specific group and then an intelligent technique detects the "neighbors" of the new user. In [17] a deep neural network extracts the content features of items and a modified, already existing, CF model takes these features into the prediction of ratings for cold-start items.

Bi-clustering has also been used in the cold-start recommendation problem. In [18], the recommendation approach consists of three phases, the filtering, the bi-clustering, and the prediction phase. In the filtering phase, a confidence level is employed to remove trivial ratings, i.e., ratings considered unimportant to a user, and popular products and frequent raters are kept to the user–product matrix. Then, in the bi-clustering phase, users and products are clustered simultaneously and a smoothing strategy is used to eliminate data sparsity and diversity of users' styles. In the prediction phase, the method seeks like-minded users (within the same cluster) and similar products (within the same cluster) to the ones under observation and combines the two predictions into one.

An example of the matrix factorization model is presented in [19], and it is based on a factored representation of the product-product similarity matrix. The latter is learned as the product of two low-dimensional latent factor matrices. In [20] a low-rank linear auto encoder is proposed which consists of an encoder that maps the user behavior space (e.g., a user–product matrix) into a user attribute space (side information), and a decoder that reconstructs the user behavior by the user attribute.

In [21] the authors propose hybrid recommender models that use content-based filtering and latent Dirichlet allocation (LDA)-based models. There are many more techniques that deal with the cold-start problem by combining collaborative filtering with content-based methods, including those using simultaneous co-clustering [22] and self-organizing maps.

Some works focus on the extraction of implicit information from the user's social networking activity [23,24]. In [25], the authors exploit data from social media to classify users' profiles and then make predictions with the use of machine learning methods, especially decision trees and random decision forests. In [26] the authors try to make a cross-site cold-start product recommendation. They use neural networks for users' and products' feature representations to transform them into user embeddings and then retrieve them in cold-start situations through a feature-based matrix factorization technique.

There are also approaches presented in the literature, which are not based on matrix operations or on a projection into latent spaces. For instance, in [27] generative adversarial networks, (machine learning frameworks based on neural networks and used in computer vision and natural language processing), are proposed to represent user information by combining the users' demographic information and their preferences.

There are also approaches that do not exploit implicit information. In [28] information theoretic measures, based on entropy, are used to tackle the problem of cold users. The measures are used to find a set of products and examine how effective the products are in learning profiles of cold users.

The cold-start recommendation problem, where most elements in certain rows or columns of the user–product matrix are 0, can be viewed as a special instance of the sparsity problem, where the user–product matrix can be extremely sparse since both the number of users and the number of products are large. The sparsity problem can be tackled by statistical techniques such as principal component analysis [29]. Empirical studies indicate that dimensionality reduction can improve recommendation quality significantly in some applications but performs poorly in others [30] since potentially useful information might be lost during this reduction process. In [31], instead of reducing the dimension of the user–product matrix (thus, making it less sparse), it is augmented based on transitive interactions between users and products.

In addressing the cold-start problem, clustering is used to group the products and/or to group the user profiles into several clusters which in turn are used to provide user/product-content information (e.g., in [32,33]). Clusters are extracted from the view of ratings, social trust relationships, user/item current context, etc. After such clustering, a new input user (group)—product (group) matrix is derived. The key idea of applying clustering is that traditional collaborative filtering approaches can then be applied to sub-matrices, which alleviates the data sparsity problem to a large extent.

In addressing the cold-start problem, ARM is also used to expand user profiles by first extracting a set of association rules for products and then by selecting such rules for each user and for each combination of products (s)he is interested in, whose antecedents much the combination. Finally, the consequences of selected rules are added to the user profile, as in [34–38]. It is worth noting, that there are also approaches that instead of using association rules, use uncertainty rules and facts [39].

Moreover, combinations of clustering and ARM are used in the literature, in addressing the cold-start problem. One approach consists of applying ARM to expand user profiles and then applying clustering to products based on the expanded input matrix to extract a group of products (e.g., in [40]). Another approach consists of applying clustering to products and then applying ARM to each extracted group of products [41], or to the whole set of groups of products [42]. In addition, another approach concerns the combination of association rules and cluster rules (e.g., in [43]).

The proposed methodology first extracts clusters of users (not of products) and then it applies discriminative pattern mining to each group of users. A similar approach is presented in [8], however standard association frequent patterns are used instead of discriminant patterns. Additionally, a similar approach is presented in [44], however, groups of users/items are formed based on the values of a selected user/item characteristic and not based on clustering. In almost all the similar methods, several ratings provided by the users are required (*in-matrix prediction*), whereas the proposed methodology provides valuable recommendations (see Section 4) to cold users that have no rating history at all (*out-of-matrix prediction*).

## 3. Proposed Methodology

In this work, a methodology is proposed to tackle the Cold-start recommendation problem, as an instance of the sparsity problem, including the extreme case of the *out-of-matrix prediction*, i.e., the recommender system must recommend products to a new user (or to detect users that might be interested in a new product). Most of the related work concerns the *in-matrix prediction*, where there are a few transactions related to new users. The proposed methodology tries to tackle the *out-of-matrix prediction* problem as well and at the same time to handle the general sparsity problem by increasing the density of the initial user–product matrix.

The pseudo-code of the proposed methodology (Algorithm 1) along with an illustration of the corresponding steps (Figure 1) are shown in what follows.

---

**Algorithm1: Recommendation Algorithm based on Discriminant Frequent Itemsets**

---

**Step 1:** Input

The set of users $U = \left\{ u_1, \ldots, u_{nofu} \right\}$, where $nofu$ is the number of users

The set of products $P = \left\{ p_1, \ldots, p_{nofp} \right\}$, where $nofp$ is the number of products

The $(nofu \times c)$ user matrix

$$UM = \{(u_i, u_{i1}, \ldots, u_{ic}) | u_{i1}, \ldots, u_{ic} \text{ are the characteristics of user } u_i \in U\}$$

The $(nofu \times nofp)$ user–product matrix $UP = \left\{ \left( u_i, \ p_1, \ldots, p_{nofp} \right) \middle| u_i \in U, \ p_1, \ldots, p_{nofp} \in P \right\}$

**Step 2:** Extract $C = \{c_1, \ldots, c_n\}$ clusters of users

**Step 3:**

For each cluster $c_i \in C$ do

    Form the $|c_i| \times nofp$ sub user–product binary matrix

$$UP_{c_i} = \left\{ \left( u_i, p_1, \ldots, p_{nofp} \right) \middle| \left( u_i, p_1, \ldots, p_{nofp} \right) \in UP, u_i \in c_i \right\}$$

    Extract frequent itemsets $FIS_{c_i} = \{ fis_{c_i 1}, \ldots, fis_{c_i m} \}$ of $UP_{c_i}$

End For

**Step 4:**

For each cluster $c_i \in C$ do

    Form the $|c_i| \times nofp$ sub user–product binary matrix

$$UP_{c_i} = \left\{ \left( u_i, p_1, \ldots, p_{nofp} \right) \middle| \left( u_i, p_1, \ldots, p_{nofp} \right) \in UP, u_i \in c_i \right\}$$

    Extract discriminant frequent itemsets $DFIS_{c_i} = \{ dfis_{c_i 1}, \ldots, dfis_{c_i q} \}$ of $UP_{c_i}$,

    where $DFIS_{c_i} \subseteq FIS_{c_i}$

End For

**Step 5:**

For each $u_i \in U$ do

    If $\left( u_i, \ p_1, \ldots, p_{nofp} \right) \in UP, p_1 = p_{2=} \ldots = p_{nofp} = 0$ * cold user

    Then

        For each $dfis_{c_i k} \in DFIS_{c_i} \wedge u_i \in c_i$ do

        $p_s = 1, s \in dfis_{c_i k}$

        End For

    Else * sparsity

        For each $dfis_{c_i k} \in DFIS_{c_i} \wedge u_i \in c_i$ do

            If $\left| \left\{ p_s \mid s \in dfis_{c_i k} \wedge p_s = 1 \right\} \right| > ct$

                If $s \in dfis_{c_i k} \wedge p_s = NULL$ Then $p_s = 1$ End If

            End If

        End For

    End If

End For

---

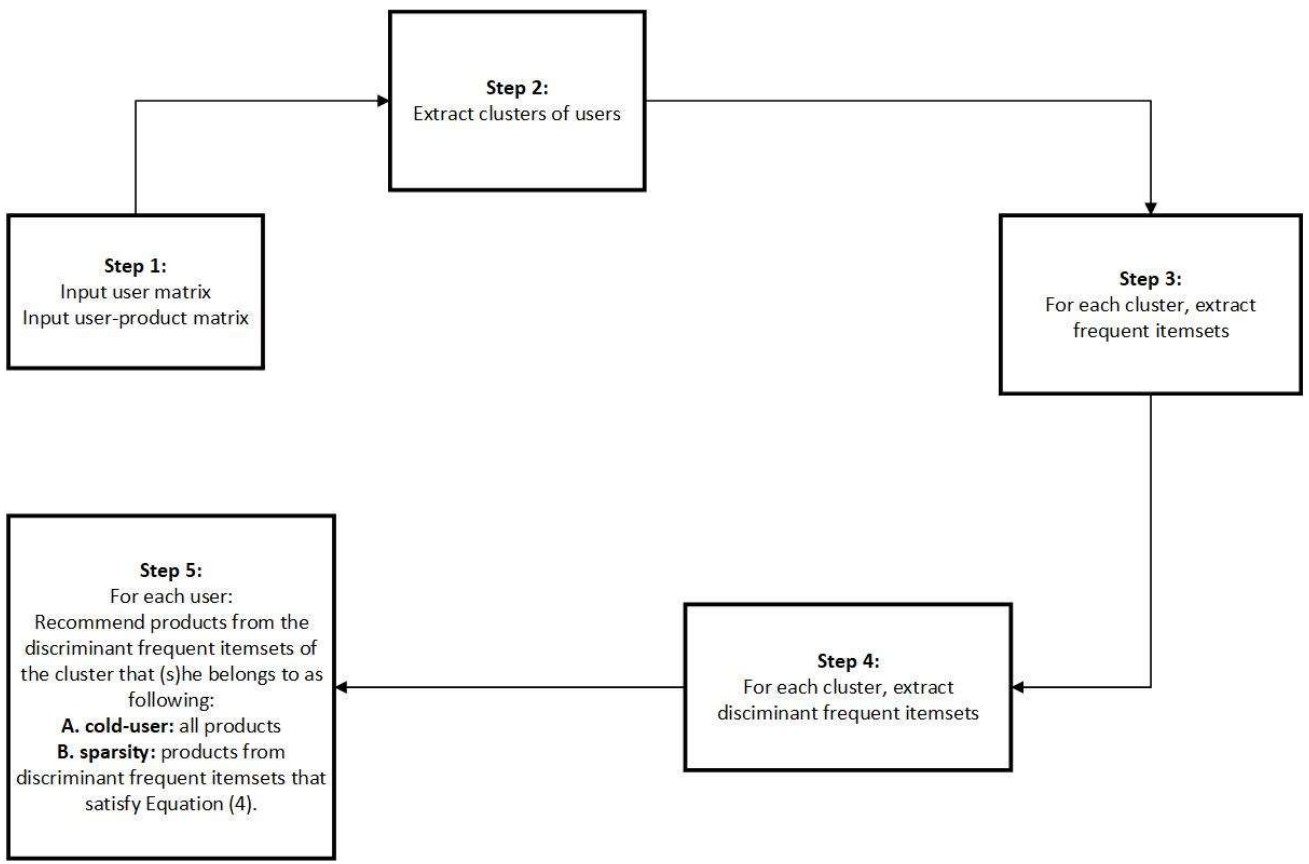

**Figure 1.** Proposed methodology.

### 3.1. Input Matrices

The input to the proposed methodology is two matrices: the two-dimensional user matrix $UM[nofu, nofuc]$ containing $nofu$ users along with their $nofuc$ characteristics (e.g., Table 1) and the user–product binary matrix $UP[nofu, nofp]$ (e.g., Table 2) indicating whether a user has rated positively (value = 1) or negatively (value = 0) a product. Given that columns represent products and rows represent users, the user–product matrix may contain blank cells (e.g., in Table 2, the cell $\left[u_{nofu}, p_3\right]$) indicating that the user $u_{nofu}$ has not provided a rating for the product $p_3$ (sparsity problem), as well as blank (*user $u_2$*) or almost blank rows $\left(user\ u_{nofu-1}\right)$ (cold case problem).

**Table 1.** User matrix UM.

|  | Age | Sex | Occupation | Country | Wage (In $) |
|---|---|---|---|---|---|
| $u_1$ | **25–35** | **F** | **Pharmacist** | **Germany** | **3000** |
| $u_2$ | 18–24 | M | Teacher | USA | 5000 |
| $u_3$ | **36–45** | **F** | **Economist** | **France** | **2500** |
| $u_4$ | **25–35** | **F** | **Doctor** | **UK** | **3000** |
| $u_5$ | 46–55 | M | Engineer | USA | 8000 |
| $u_6$ | 18–24 | M | Professor | USA | 6000 |
| $u_7$ | 25–35 | M | Lawyer | USA | 8000 |
| $u_8$ | 18–24 | F | Teacher | USA | 6000 |
| $u_{nofu-1}$ | Over 55 | F | Teacher | Canada | 6000 |
| $u_{nofu}$ | 46–55 | M | Assistant | China | 1500 |

**Table 2.** User–product matrix UP.

|  | $p_1$ | $p_2$ | $p_3$ | $p_4$ | $p_5$ | $p_{nofp-1}$ | $p_{nofp}$ |  |
|---|---|---|---|---|---|---|---|---|
| $u_1$ | 1 | 0 | 1 |  |  |  |  | Not updated |
| $u_2$ | 1 |  |  | 1 | 1 |  |  | *Updated (\*cold user)* |
| $u_3$ | 1 | 1 | 1 |  | 1 |  |  | Updated |
| $u_4$ | 1 |  | 0 | 1 | 0 |  | 1 | Not updated |
| $u_5$ | 0 | 1 | 0 |  |  |  | 0 |  |
| $u_6$ | 1 |  | 0 |  |  |  | 1 | *Not updated* |
| $u_7$ | 1 |  | 1 | 1 |  |  |  | *Not updated* |
| $u_8$ | 1 |  | 1 | 1 |  |  |  | *Updated* |
| $u_{nofu-1}$ |  |  |  | 0 |  | 1 |  |  |
| $u_{nofu}$ | 0 | 1 |  | 0 | 1 |  |  |  |

### 3.2. Clustering Procedure

Initially, during a preprocessing phase, users are clustered to certain clusters by applying a clustering algorithm to the user matrix. The selection of the clustering algorithm depends on the type of users' characteristics. Assume that a set $C$ that contains $n$ *clusters of users* $(c_1, \ldots, c_n)$ is extracted at the first phase.

### 3.3. Extraction of Frequent Itemsets

Then, during Step 3, for each cluster $c_i \in C$, an ARM algorithm is used to extract the frequent itemsets along with their frequencies. When the ARM algorithm is applied to the two-dimensional binary matrix, it extracts the frequent itemsets by taking as input for each row (i.e., transaction) the columns (i.e., items) containing the value 1, i.e., the products that the user purchased.

The ARM algorithm is used to extract for each cluster $c_i \in C$, the set of frequent itemsets $FIS_{c_i}$. The choice of the ARM algorithm is not critical, and it depends on the size of the user–product matrix. For the cluster $c_i \in C$, $1 \leq i \leq n$, the relative frequency of each frequent itemset $fis_{c_i a} \in FIS_{c_i}$ is:

$$RFfis_{c_i a} = Ffis_{c_i a}/|c_i| \tag{1}$$

where $c_i \in C$, $1 \leq i \leq n$ and $fis_{c_i a} \in FIS_{c_i}$, $1 \leq a \leq m$, where $m$ is the total number of frequent itemsets for the cluster $c_i$.

The relative frequency of an itemset represents how frequent this itemset is, in all the transactions, i.e., the number of transactions that contain this itemset. In the proposed methodology, the relative frequency is the total number of users (rows) that the cluster $c_i$ contains, where $c_i \subseteq U$.

### 3.4. Extraction of Discriminant Itemsets

During Step 4, the discriminant itemsets are extracted for each user cluster. For each extracted frequent itemset $fis_{c_i a}$, it is checked if it is included in other user clusters. Given that $\theta > 1$ and $0 < \varphi < \frac{1}{\theta}$ are user-specified thresholds, a frequent itemset $fis_{c_1 1}$ is discriminant in the cluster $c_1$, iff the following hold [45]:

$$\left(n * RFfis_{c_1 1}/ \sum_{i=1}^{n} RFfis_{c_i 1}\right) \geq \theta \tag{2}$$

$$RFfis_{c_1 1} \geq \varphi * \theta, \text{ where } 0 < \varphi < \frac{1}{\theta} \tag{3}$$

The use of these thresholds ensures that itemsets with extremely low frequencies are eliminated.

### 3.5. Recommendation

During Step 5, the user–product matrix is updated based on the extracted discriminant itemsets. Each row representing a certain user is updated based on discriminant itemsets of the user cluster it belongs to. A discriminant itemset $dis_{c_i a}$ is used for updating a row if it conflicts with the latter for less than a predefined threshold $ct$.

$$\text{Discriminant itemset } dis_{c_i a} = \{p_i \mid 1 \leq i \leq nofp\} \in DFIS_{c_i} \text{ conflicts with a}$$
$$row_i = \left\{cell_1, \ldots, cell_{nofp}\right\}, \textbf{\textit{iff}} \; |\{cell_i | cell_i = 0 \text{ and } p_i = 1\}| < \; |dis_{c_i a}| * ct \tag{4}$$

Intuitively, a discriminant itemset that recommends to a user to purchase a considerable (above threshold) number of products ($p_i = 1$) contrary to explicitly stated user's preference ($cell_i = 0$), cannot be trusted.

Otherwise, if a discriminant itemset can be trusted then the row is updated by setting ($cell_i = p_i$), unless $cell_i = 0$.

### 3.6. An Illustrative Example

The following example depicts the proposed methodology in a more analytic way. Assume that one of the extracted clusters $c_1$ contains the users $u_1$, $u_3$, $u_4$. Assume that a discriminant itemset extracted is $\{p_1, p_3\}$. User $u_1$ has positively rated $p_1$ ($cell_1 = 1$) and $p_3$ ($cell_3 = 1$), thus the corresponding row is not actually updated. User $u_3$ has positively rated $p_1$ ($cell_1 = 1$), while (s)he did not rate $p_3$ ($cell_3 = null$), thus the corresponding row is updated by setting ($cell_3 = 1$).

User $u_4$ has positively rated $p_1$ ($cell_1 = 1$) and negatively $p_3$ ($cell_3 = 0$), thus the corresponding row is not updated since the method does not change the explicitly stated user's preferences.

The extracted cluster $c_2$ contains users $u_2$, $u_6, u_7$, $u_8$. Assume that a discriminant itemset for that cluster is $\{p_1, p_4, p_5\}$. User $u_6$ has positively rated $p_1$ ($cell_1 = 1$) and negatively $p_4$ ($cell_4 = 0$), while (s)he did not rate $p_5$ ($cell_5 = null$). Given that the threshold $ct = 0.25$, the corresponding row is not updated by setting ($cell_5 = 1$) since the discriminant itemset $\{p_1, p_4, p_5\}$ conflicts with the row for more than the predefined threshold ($1/3 > 0.25$). Note that if the threshold was set to $ct = 0.5$, the corresponding row would have been updated by setting $cell_5 = 1$ ($1/3 < 0.5$). User $u_7$ has already positively rated $p_1$ ($cell_1 = 1$), $p_4$ ($cell_4 = 1$), $p_5$($cell_5 = 1$), thus the corresponding row $u_7$ is not updated. User $u_8$ has positively rated $p_1$ ($cell_1 = 1$) and $p_4$ ($cell_4 = 1$) while (s)he did not rate $p_5$ ($cell_5 = null$), thus the corresponding row is updated by setting ($cell_5 = 1$). At last, user $u_2$ is a cold user with no rating history, thus the discriminant itemset updates $cell_1$, $cell_4, cell_5$ with value 1.

## 4. Empirical Results

In this section, we evaluate the proposed methodology using the MovieLens dataset [46] which is widely used as a benchmark in the recommender systems' field [8–14]. We will first describe the input dataset and the experimental setup and then we will present the experimental results.

The MovieLens dataset consists of 100,000 ratings (from 1 to 5) from 943 users on 1682 movies. Each user has rated at least 20 movies. Simple demographic info for the users (age, gender, occupation, zip) is used.

The proposed methodology takes into consideration users' attributes and creates clusters of users that are similar. Then, it exploits these similarities to recommend products to users with small or no purchase (or rating) history (cold-start users).

At first, we applied the Kmodes algorithm [47] to perform user clustering, since the user's demographic attributes are categorical. Then, we transformed the ratings within the

MovieLens dataset into binary values (1 for ratings higher than 2 and 0 otherwise) to apply the ARM algorithm. Thus, the frequent itemsets for each cluster of users were extracted by applying the Apriori ARM algorithm [48].

In the case of a user cluster, a discriminant itemset consists of products that can be recommended (cell value equals 1) to each user of a cluster, based on Equation (4).

In the MovieLens dataset, the user–product matrix contains user IDs and movie IDs. The initial data matrix with 943 rows and 1682 columns, contains 100,000 ratings, which corresponds to a density of around 6.3%. The user–product matrix is too sparse, i.e., includes users that have not rated a lot of movies. The objectives of the proposed methodology are:

- To increase the density of the initial matrix by recommending more movies to users who have a small rating history (*in-matrix prediction*);
- To recommend movies to *new users* with no rating history at all, based on the rating history of users with similar characteristics (*out-of-matrix prediction*).

To validate the experimental results, we used the *k-fold* cross-validation procedure. In this type of validation, a data set is first randomly divided into *k* disjoint folds that have approximately the same number of instances. Then, every fold in turn plays the role of testing the model induced from the other *k-1* folds [49].

The rows (users) of the initial matrix were randomly divided into $k = 10$ disjoint subsets with each one consisting of $m_i = nofu/k = 94$ rows, where $nofu = 943$ is the total number of rows (users) and $i \in [1, 10]$. A *10-fold* cross-validation is performed for *out-of-matrix prediction*, i.e., all the rows of a subset are removed from the initial input matrix, and they are considered as the test set, thus representing the cold (new) users with no rating history for products. In addition, a *10-fold* cross-validation is performed for *in-matrix prediction*, i.e., for each row of a subset, a randomly selected fraction *p* of cell values equal to 1 or 0 are deleted, thus representing the cold (new) users with a little rating history for products. The deleted cells of each row of a subset (not the entire row) form the test set and *p* varies from 30% to 90% of each row's number of cells. The cells were removed by randomly selecting $p/2$ of cells of a row with value 1 and $p/2$ of cells of a row with value 0.

To evaluate the proposed methodology, we compare the added cell values within the test set to the initial values of these cells within the initial user–product matrix, i.e., we test how well the proposed methodology predicts if a new user might like a product. We adopt well-known literature evaluation measures that are used in similar problems (e.g., in diagnostic tests that identify if a disease is present compared to definitive tests that show the true prevalence of a [50–52] disease). These measures are sensitivity and positive predictive value. In this work, the sensitivity measure is defined as the proportion of the true positives, i.e., the number of users successfully predicted by the proposed methodology as advocates of a product (predicted cell value is 1) to the number of all positives, i.e., the number of users who have explicitly stated that they like a product within the initial user–product matrix. In addition, in this work, the positive predictive value of a test is defined as the proportion of the true positives to both true and false positives, i.e., the number of users predicted (successfully or not) by the proposed methodology as advocates of a product (predicted cell value is 1).

Thus, the following equations represent the adopted measures:

$$Sensitivity\ (Recall) = \frac{TP}{P} \tag{5}$$

$$Positive\ Predictive\ Value - PPV\ (Precision) = \frac{TP}{TP + FP} \tag{6}$$

$$F_1\ measure = 2 * \frac{Precision * Recall}{Precision + Recall} \tag{7}$$

where $TP$, $FP$ is the true and false positives, respectively (predicted cell value = 1), and $P$ is the number of all positives in the initial user–product matrix (existed cell value = 1). Obviously, the better achieved results are indicated by a value approaching 1 for both measures.

Note that there is a difference in calculating the measure of sensitivity within this paper compared to other similar problems (e.g., in disease diagnosis), because of the existence of empty cells (null cell value). Since the proposed methodology, apart from adding positives (predicted cell value = 1), is allowed to leave empty cells within the test subset, the sensitivity measure could be low. This is because some positives in the initial input matrix (existing cell value = 1) can be left empty, which, although they are not false positives, cannot be considered true positives, as well.

To evaluate the proposed methodology for the sparsity problem, we introduce three new measures. The plentifulness measure indicates how many initially empty cells were filled with respect to existing positives and it is defined as the proportion of added positives (predicted cell value is 1) into initially empty cells to the number of initial positives. The plentifulness rate measure indicates how many initially empty cells were filled with respect to all existing empty ones and it is defined as the proportion of added positives into initially empty cells to the number of initially empty cells. The plentifulness predictive value measure indicates how many initially empty cells were filled with respect to both the latter cells and the initially positives that were left empty and it is defined as the proportion of added positives into initially empty cells to the sum of the latter and the cells left empty while they were initially positive. Thus:

$$Plentifulness = \frac{NP}{P} \tag{8}$$

$$Plentifulness\ rate = \frac{NP}{E} \tag{9}$$

$$Plentifulness\ Predictive\ Value = PLPV = \frac{NP}{NP + EP} \tag{10}$$

where $E$ is the number of empty cells (null cell value) in the initial user–product matrix; $NP$ is the number of added positives (predicted cell value is 1) into previously empty cells and $EP$ is the number of cells left empty while they were positives in the initial user–product matrix (existed cell value = 1). Obviously, the better achieved results are indicated by a value approaching 1 for all measures.

We implemented the proposed methodology using Python 3.11 and all the experiments were performed on a common Personal Computer Intel Core TM i7-10750 CPU (2.59 GHz) with 16 GB RAM. The time complexity of the proposed methodology is dominated by the runtime of the Apriori algorithm (e.g., 37.65 s for minimum support set to 15% and 1.97 for minimum support set to 20%).

Figures 2–6 present the experimental results obtained by the 10-fold cross-validation procedure. More specifically, Figures 2 and 3 show the *out-of-matrix prediction* results obtained by the proposed methodology, since all the rows (users) included in the test set were cleared and then they were updated by the proposed methodology. In Figure 2, the cross-validation results (average over the $k = 10$ runs) for the evaluation measures are shown with respect to different values of $\theta$ parameter while the minimum support is set to 15% and $\varphi$ parameter to 0.001 (see Equations (2) and (3)). In Figure 3, the cross-validation results (average over the $k = 10$ runs) for the evaluation measures are shown with respect to different values of minimum support while the $\theta$ parameter is set to 1 and $\varphi$ parameter to 0.001. Note that a higher value of minimum support reduces the number of extracted frequent itemsets and hence that of discriminant itemsets. Therefore, if there are not sufficient extracted (discriminant) itemsets (i.e., high minimum support) the accuracy of the methodology is decreased. However, there is a wide range of minimum support (0–25%) which guarantees the very high accuracy of the proposed methodology. The value

of $\varphi$ parameter does not affect remarkably the performance while the value of $\theta$ parameter affects only slightly the performance.

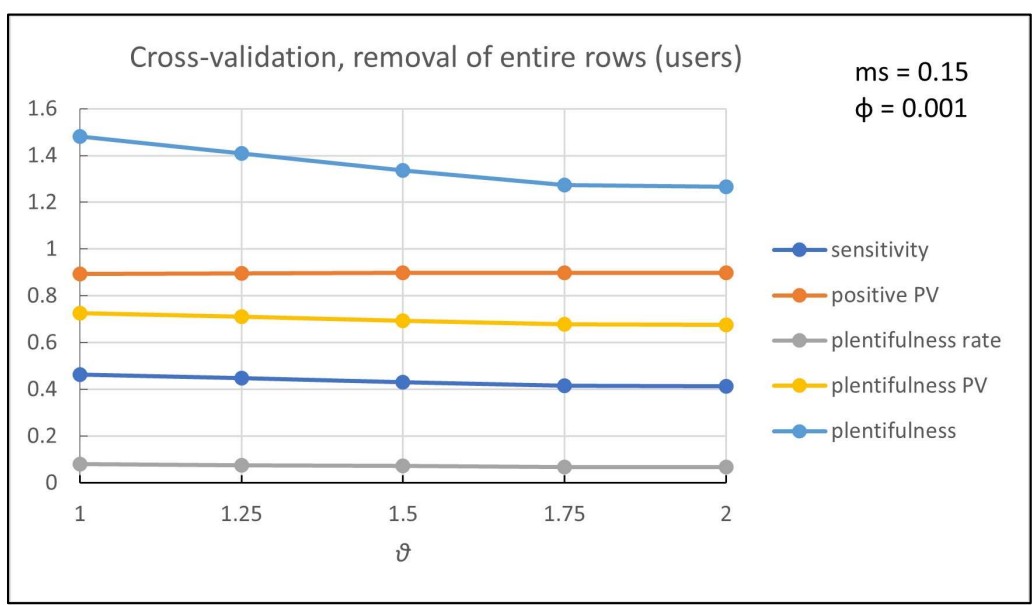

**Figure 2.** Cross-validation results (100% removal, minimum support = 0.15).

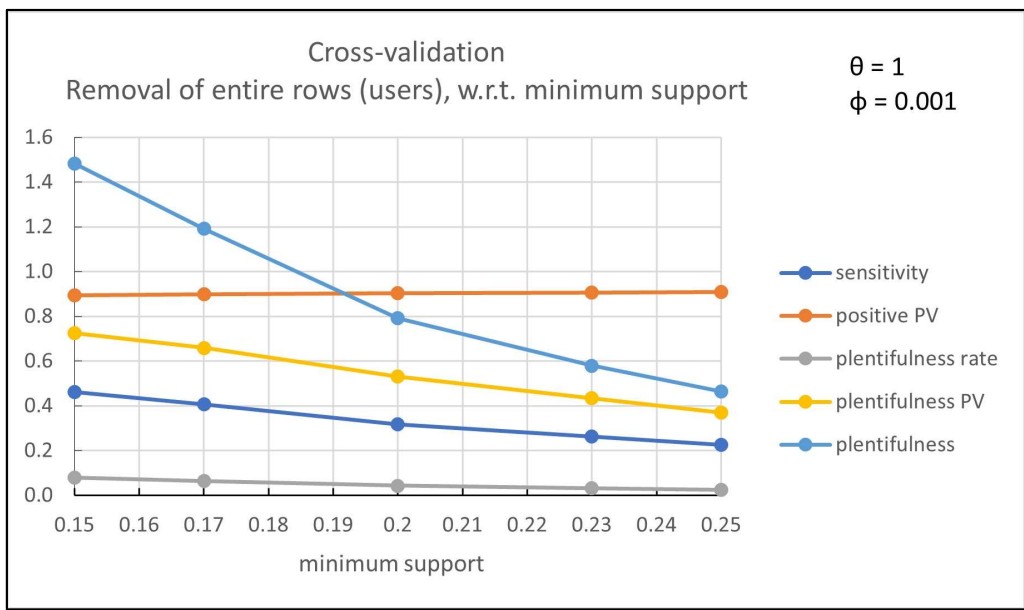

**Figure 3.** Cross-validation results (100% removal, with respect to minimum support).

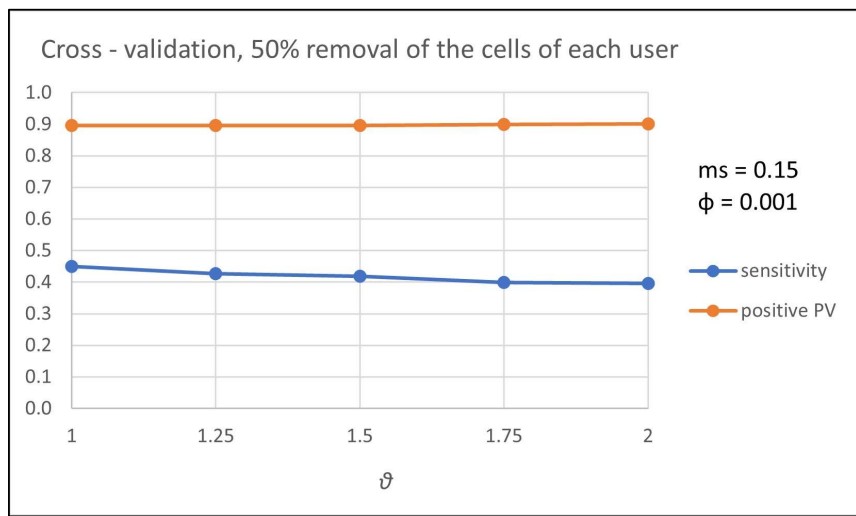

**Figure 4.** Cross-validation results (50% removal, minimum support = 0.15).

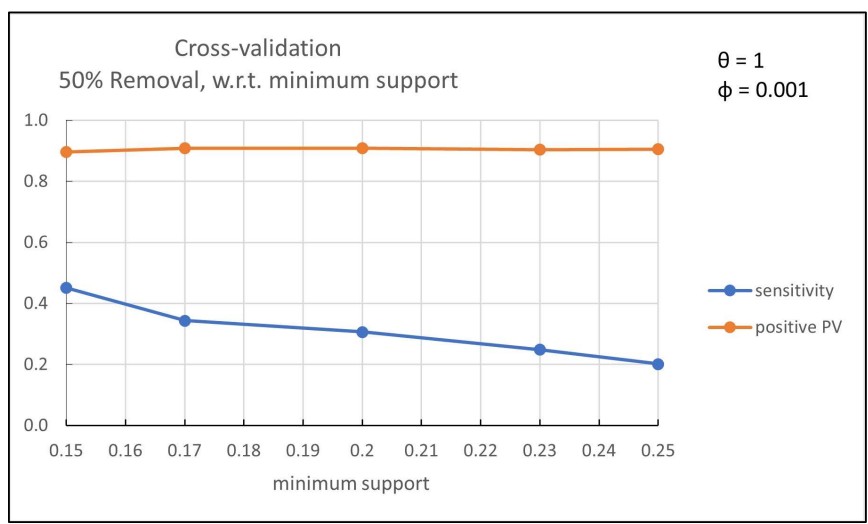

**Figure 5.** Cross-validation results (50% removal, with respect to minimum support).

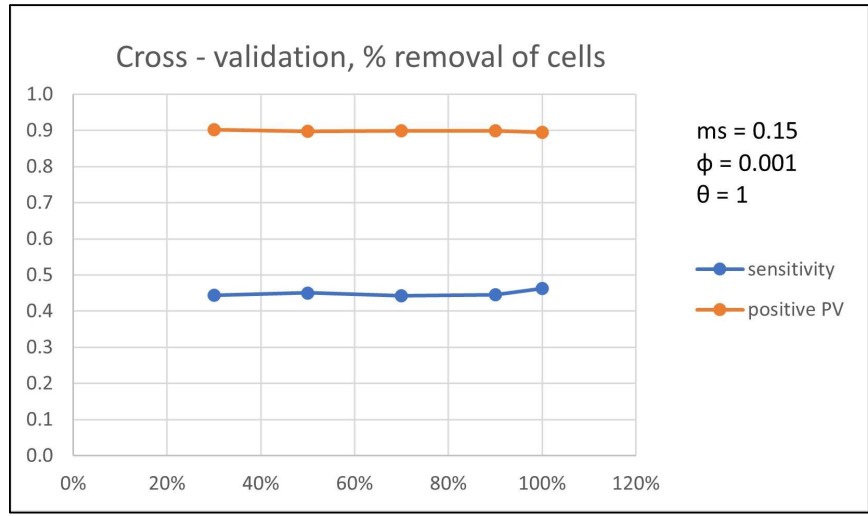

**Figure 6.** Cross-validation results (with respect to remove percentage, minimum support = 0.15).

The results in Figure 2 show that the value of plentifulness is greater than 1, which means that the proposed methodology recommends more products than those included in the initial user–product input matrix. This is a very important result as far as the sparsity problem is concerned. Note that the density of the initial user–product matrix is 6.3%. which is increased to about 14% by the proposed methodology (plentifulness rate $\approx$ 8%), while only a small fraction of initially non-empty cells turned into empty cells by the proposed methodology (plentifulness predictive value $\approx$ 0.73). At the same time, the results in Figure 2 show that the value of positive predictive value is close to 1, which means that the proposed methodology does not recommend products that were negatively rated by the user (FP), although some of the products that were initially positively rated by the user are not recommended (sensitivity $\approx$ 0.5).

Therefore, the proposed methodology tackles the sparsity problem by recommending more new products than the known ones while, although it cannot detect all the known ones, it does not recommend products that the users do not like, based on the initial user–product input matrix.

The above results hold also for the *in-matrix prediction* setup, where 50% of the non-empty cells of each row (user) included in the test set were cleared and then they were updated by the proposed methodology. Therefore, only the sensitivity and positive predictive value measures are considered. In Figure 4, the cross-validation results (average over the $k = 10$ runs) for these evaluation measures are shown with respect to different values of $\theta$ parameter while the minimum support is set to 15% and $\varphi$ parameter to 0.001. In Figure 5, the cross-validation results for the evaluation measures are shown with respect to different values of minimum support while the $\theta$ parameter is set to 1 and $\varphi$ parameter to 0.001. Note that the fraction of cells of a row that is cleared does not affect the performance of the proposed methodology. In Figure 6, the cross-validation results (average over the $k = 10$ runs) for these evaluation measures are shown with respect to different fractions of cleared cells, while $\theta$ parameter is set to 1, minimum support to 15%, and $\varphi$ parameter to 0.001.

Therefore, based on the experimental results, the proposed methodology can efficiently handle both the *in-matrix* and *out-of-matrix prediction*, i.e., both matrices that are sparse and matrices that contain users even with no rating history at all.

*Comparison to Similar Methods*

To further evaluate the accuracy of the proposed methodology, it is compared to similar methods presented in the literature, with respect to precision (positive predictive value), recall (sensitivity), and F1 measures. Table 3 shows the experimental results for these three measures concerning the MovieLens dataset [46]. The results shown for the proposed methodology concern both the *out-of-matrix* and the *in-matrix prediction*, while those for similar methods only the *in-matrix prediction*. The proposed methodology outperforms similar methods with respect to all the measures, even for the *out-of-matrix prediction* setup.

**Table 3.** Comparison to similar methods.

| Author | Precision | Recall | F1-Measure |
|---|---|---|---|
| **Proposed method (*out-of-matrix*)** | **0.895** | **0.463** | **0.61** |
| **Proposed method (*in-matrix*, 30–90%)** | **0.903–0.900** | **0.444–0.445** | **0.595–0.595** |
| de Carvalho et al. (2020) [8] | 0.035 | 0.121 | 0.052 |
| Yanxiang, L. et al. (2013) [9] | 0.651 | - | - |
| Peng, Lu et al. (2016) [11] | 0.786 | 0.287 | 0.367 |
| Park and Chu (2009) [12] | 0.666 | 0.239 | 0.311 |
| Bobadilla, Ortega [13] | 0.37–0.57 | 0.25–1 | - |
| Huang and Yin (2010) [14] | - | - | 0.308 |

## 5. Discussion

We presented a methodology that can handle efficiently sparsity problems in recommender systems and especially can provide product recommendations to cold-start users, i.e., new users with no rating history. We addressed the sparsity and cold-start problems by applying a combination of clustering and association rule mining algorithms.

First, we apply a clustering algorithm to the set of users based on their characteristics. The choice of the clustering algorithm is not critical, and it depends on the type of users' characteristics. However, algorithms capable of identifying the number of clusters are preferred (e.g., [53]). It is worth noting that, the used clustering algorithm can provide more solid groups of users with similar purchasing behavior if it is applied to implicit data deriving from users' social media activity or profiles.

Then, we apply discriminative itemset mining that extracts discriminant frequent patterns for each different user cluster. The use of discriminative itemset mining is an innovation in addressing the sparsity and cold-start problems.

It is important to note that the proposed methodology can straightforwardly be applied also to predict the products that the user does not like. This can be achieved simply by extracting frequent itemsets and hence discriminant itemsets based on cells with cell values equal to 0. In this case, a discriminant itemset consists of products that cannot be recommended to each user of a cluster, based also on Equation (4). Then, the proposed methodology can update the initial user–product input matrix by adding zero to certain cells whether they were initially empty or not.

Additionally, it is very important to note that the proposed methodology can be straightforwardly applied also to the recommendation of newly released products with no purchase history (cold-start products) to users that have bought (or rated) similar products in the past. At first, the clustering procedure is applied to a product matrix of product characteristics and then the ARM algorithm and the extraction of discriminant frequent itemsets are performed to the subsets of products. Finally, the initial user–product matrix is updated by adding 1 (recommended products) or 0 (not recommended products) cell values for those products that have never or have scarcely been bought in the past.

The performed empirical tests on large-scale data (MovieLens dataset) show that the proposed methodology tackles both the sparsity problem and the *in-matrix* and *out-of-matrix prediction* setup of the cold-start problem by recommending more new products than the known ones, while at the same time not recommending products that the users do not like.

## 6. Conclusions

In this paper, we proposed a methodology that can handle the cold-start recommendation problem. This method can be applied to sparse user–product matrices which contain users with small or no rating history at all. At first, the proposed methodology considers users' characteristics and performs a clustering procedure to create groups of similar users. Then, by using the rating history of the users who belong to the same cluster, the methodology extracts the frequent itemsets, i.e., combinations of products that the users have rated positively. The discriminant frequent itemsets of each cluster are extracted afterward, i.e., itemsets that are highly frequent in a cluster of users, but less frequent or infrequent in other clusters of users. The initial user–product matrix is enriched by adding to each user's empty cells the positive ratings of the products based on the discriminant itemsets of the cluster the user belongs to.

The proposed methodology is flexible enough and can also be applied to predict products that the user does not like or recommend products that are newly released, and that no purchase history exists.

The presented empirical results demonstrate the efficiency and accuracy of the proposed methodology, which, to our knowledge, exhibits the highest precision (0.903) among all the similar methods presented in the literature addressing the cold-start problem (See Table 3).

### 6.1. Limitations

The proposed methodology relies on the quality of the clustering procedure. Obviously, the higher the clustering quality, the better the performance of the proposed methodology. It is known that similar users can only be selected from the fixed size of cluster members, and in general, a fewer number of similar users can be identified compared with the whole space, thus clustering-based methods still suffer from relatively low accuracy [54].

However, the presented empirical results are achieved by using only simple demographic info for the users (age, gender, occupation, zip). It seems that the performance of the proposed methodology will further increase by using more informative characteristics of the user's/product's behavior and context (e.g., social network information, social tags, etc.). Moreover, the clustering quality can be increased by applying more dedication to each application technique such as biclustering, text categorization, fuzzy *k*-means, etc.

In addition, the minimum support of extracted discriminant frequent itemsets has a great impact on recommendation quality. It is known that such methodologies suffer from the problem of low support.

### 6.2. Future Work

In this work, we focused on recommending products that the users might like, since this is the usual goal of CF. Additionally, it has been shown that recommending products that the users might dislike is not quite effective by using standard association rule mining [36]. We are currently investigating the efficiency of the proposed methodology for products that the user dislikes.

We are also investigating the efficiency of the proposed methodology to cold-start products, i.e., new products with no purchase history.

Moreover, we are investigating the improvement of the performance of the proposed methodology by applying clustering to implicit user data deriving from users' social media activity or profiles.

**Author Contributions:** Conceptualization, B.B. and A.P.; methodology, B.B.; validation, B.B. and A.P.; resources, A.P.; writing, B.B. and A.P.; visualization, A.P. All authors have read and agreed to the published version of the manuscript.

**Funding:** This research received no external funding.

**Data Availability Statement:** Publicly available datasets were analyzed in this study. This data can be found here: https://grouplens.org/datasets/movielens/100k/ accessed on 15 December 2022.

**Acknowledgments:** The authors would like to thank Anastasios Tsimakis for his invaluable support during the implementation of the proposed method.

**Conflicts of Interest:** The authors declare no conflict of interest.

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
