# Peer review of "Addressing the Cold-Start Problem in Recommender Systems Based on Frequent Patterns"

_algorithms, doi:10.3390/a16040182_

Round 1

Reviewer 1 Report

Greetings,

The paper is well written, but it is necessary to make certain corrections in the paper. It is necessary to make certain changes in the introduction to better define the goals, then to emphasize the contributions of this paper and to say which gaps this paper solves. Selection 2. Related work should be renamed to Selection 2 Literature review. Selection 5. Empirical results should be 4. Empirical results. It is necessary to separate the discussion from the conclusion. In the conclusion, limitations should be stated and directions for future research should be given.

Regarding the main question of the research, I said that it is necessary to explain in more detail, what is the goal of the research and what is its contribution and what gaps does this research solve. Then it is necessary to state it primarily in the introduction of this paper.
The topic of this paper is suitable for Algorithms journals, but the paper needs to be further refined in order to be accepted. I stated that it must be emphasized in the introduction which research gap this paper addresses and what the contribution is.
This paper does not provide any new methods, ways of using, etc., but provides a new methodology for solving these research problems.
With the research methodology, it is necessary to explain these steps, which are listed in Figure 1.
The discussion and conclusion were put together, and I suggested that these selections be separated and that the conclusion explain the limitations of the paper and give directions for future research.
Tables 1 and 2 need to be slightly edited to make them more suitable for journals. It is also necessary to unify expressions of different sizes, etc.
The references are appropriate and it would be desirable to mention a few more recent references that solve similar research problems through the discussion.

All the best.

Reviewer 2 Report

The idea in this paper is to predict empty entries in a user-item matrix for addressing the cold start problem in recommender systems. This is achieved by clustering and completing void values with novel preferences for the entries, calculated based on dissimilarity with frequent values. 

While the idea is appealing and is correctly explained and motivated, there are some issues that need to be improved in the paper in order to make it acceptable for publication:

1) The specification of the algorithm (equations (1)-(4)) is rather cumbersome in notation, and not very precise. Please, simplify notation and explain clearly the meaning of the equations and steps in order to highlight the interpretation and clarity of the algorithm.

2) The plots with the experimental results contain only a few points for each metric. Reconsider if this is the best form of presentation.

3) In all the experimental tests shown in the paper, the value of positive PV is practically 0.9, irrespective of other parameters. Please, explain this fact.

4) Many entries in the bibliography lack sufficient information. Please, review and fix the citations.

Reviewer 3 Report

The paper is focused on the use of discriminative itemsets for facing the traditional cold-start problem in recommender systems. Even though cold start has been a well-explored problem in RS, the work is very interesting and promising regarding its aim is the exploitation of discriminative patterns in such a context.

Below we present some directions that should be followed by the authors for improving the quality of presented work, as possible publication in the journal.

-The novelty of the current work should be explicitly highlighted at the end of the Introduction section, in relation to previous research. A comparative table could also help in this direction.

-The presentation of the methodology (Section 3) should be improved. We suggest to structure this section in subsections, with a better explanation of the four stages of the proposal. Furthermore, we suggest including an algorithm, using the usual algorithm notation, illustrating the whole proposal.

-The illustrative example would be very useful; however, we think the behavior of the proposal with new users (cold-start users) should be shown better. In Table 2, it is not clear which is the profile of the new user.

-In the experimental analysis, it is expected the comparison with previous works.

-Even though the literature analysis was pretty appropriate, we suggest to include some pioneer works on association rule mining-based cold-start recommendation, such as:

*Lin, W., Alvarez, S. A., & Ruiz, C. (2002). Efficient adaptive-support association rule mining for recommender systems. Data mining and knowledge discovery, 6, 83-105.

*Leung, C. W. K., Chan, S. C. F., & Chung, F. L. (2008). An empirical study of a cross-level association rule mining approach to cold-start recommendations. Knowledge-Based Systems, 21(7), 515-529.

-The format of the references needs to be corrected, regarding that in most of cases the names of the journals or the conferences are missing.

We suggest Major Revision.

Round 2

Reviewer 1 Report

Greetings,

The authors have respected the reviewers' comments. The paper should now be accepted.

All the best.

Reviewer 2 Report

The authors have completed the gaps and formulated the algorithm in precise terms, which were my previous main concerns. The results presented in the paper demonstrate a significant improvement over other similar techniques. The general presentation of the work is correct. Therefore, I would recommend publication of this manuscript.

Reviewer 3 Report

I have carefully checked the new version of the paper. The authors have covered all my previous concerns related to remark the novelty of the paper, the screening an algorithm presenting the whole proposal, and the comparison with previous related works.

At this stage we think that the paper has been improved, and that the current version is appropriate for publication.

We suggest paper acceptance.